# TRPA1 Agonists and Bladder Nociception in Female Rats Suggest Potential for Nutraceutical Benefit from Cinnamon

Timothy J. Ness *[iD], Amer Babi [iD], Madeline E. Ness and Cary DeWitte

Department of Anesthesiology and Perioperative Medicine, University of Alabama at Birmingham, Birmingham, AL 35205, USA
* Correspondence: tness@uabmc.edu

**Abstract:** TRPA1-related drugs alter sensation, particularly in conditions of inflammation. To further characterize the role of these drugs in bladder sensation, the TRPA1 agonist cinnamaldehyde (CMA) and oral true cinnamon spice were examined in preclinical models of bladder pain. Female adult rats, with and without acute zymosan-induced cystitis, were anesthetized and visceromotor (VMR) and cystometric responses to urinary bladder distension (UBD) were determined following either the intravesical administration of CMA/vehicle solutions or the oral administration of true cinnamon/vehicle. ELISA measures of bladder TRPA1 content were also determined. Acute cystitis resulted in increases in bladder TRPA1 content and produced an increased vigor of the VMRs to UBD and a lowering of micturition volume thresholds for activation of a micturition response. Intravesical CMA produced a robust inhibition of VMRs to UBD in rats with cystitis but not in those without. Micturition volume thresholds were lowered by CMA in rats without cystitis but had no additional effect in rats with cystitis. Oral cinnamon also produced a robust inhibition of VMRs to UBD in rats with cystitis and a mild augmentation of VMRs to UBD in rats without cystitis. A potentially analgesic effect of the spice, true cinnamon, in the treatment of the pain of acute cystitis was suggested by these preclinical studies. Human studies are indicated.

**Keywords:** TRPA1; cinnamon; cinnamaldehyde; urinary bladder





## 1. Introduction

Transient receptor potential ankyrin 1 (TRPA1) is a non-selective cation channel activated by a variety of stimuli including cinnamaldehyde (CMA), allyl isothiocyanate (mustard oil), allicin, acrolein, $H_2S$, and noxious cold temperatures. TRPA1 is recognized as an important channel for mechanosensory, temperature, and nociceptive function and may play a unique role under conditions of inflammation [1,2]. A role for the TRPA1 receptor channel in bladder sensation was demonstrated long before identification of the receptor itself. Some of the earliest models of bladder pain (e.g., [3,4]) infused the TRPA1 agonist mustard oil into the bladder of rats to produce local neurogenic inflammatory changes and bladder hyperalgesia. Subsequent studies have identified the presence of TRPA1 receptors in both bladder epithelium and its neuronal components in humans [5] and rodents [6]. Painful bladder disorders such as interstitial cystitis in humans have been associated with increased levels of TRPA1 within the bladder [7]. Due to this information, drugs related to TRPA1 transduction have been proposed as good candidates for therapeutic interventions related to nociceptive and particularly inflammatory pain [8–11].

It is notable that most non-human animal studies demonstrating TRPA1 mechanisms associated with the bladder have utilized acute inflammation of the bladder as a co-manipulation and reported alterations in protein expression or cystometric measures as their primary outcomes (e.g., [12–14]). A few studies have utilized classic pseudaffective measures of bladder nociception such as visceromotor responses (VMRs) and cardiovascular responses to urinary bladder distension as endpoints (e.g., [15,16]). Numerous other

pain-related models such as those associated with somatic pain, pancreatitis, gastritis, endometriosis, and musculoskeletal pain have also demonstrated TRPA1 mechanisms leading, in some cases, to increased nociceptive responses and, in others, to antinociceptive mechanisms [17–23]. The only thing that is clear from a review of the literature is that our understanding of TRPA1-related mechanisms of the bladder is incomplete.

Numerous TRPA1 agonists are present in savory foods including allicin (found in garlic and onions) and cinnamaldehyde (CMA) found in cinnamon. The effect of these dietary exposures to TRPA1 agonists on ongoing pain in humans has been variable with claims of effect ranging from exacerbations of pain to analgesia. Painful bladder disorders such as interstitial cystitis/bladder pain syndrome (IC/BPS) commonly report that the ingestion of various foods produce flares in their pain [24]. Therefore, we hypothesized that ingestion of foods such as cinnamon, which have actions on TRPA1 receptors, would likely produce increased pain and so should be avoided. To test this hypothesis, we decided to perform preclinical experiments in rodent models. We first verified that bladder inflammation alters TRPA1 content within the bladder. Then, we directly infused CMA into the bladder of anesthetized rats by a transurethral intravesical route in the presence and absence of ongoing cystitis. Due to the unexpected observation of a robust antinociceptive (rather than pronociceptive) effect of CMA in rats with cystitis, we then characterized the dose-dependent effects of oral cinnamon on rat bladder nociceptive responses when the spice was administered by oral gavage.

## 2. Materials and Methods

### 2.1. General Overview

Subjects were adult (12 week) female Sprague Dawley rats (220–290 g; Source: Harlan/Invigo Laboratories, Prattville, AL, USA). These studies were approved by the University of Alabama at Birmingham Institutional Animal Care and Use Committee (IACUC-10250, app. 10.30.2014; IACUC-22026, app. 4.7.2020). Female rats were exclusively utilized due to the ease of bladder cannulation and the much higher female incidence of cystitis. The studies consisted of three different groups of experiments: 1. an assessment of the effect of acute zymosan-induced bladder inflammation on TRPA1 protein content in the bladder; 2. a controlled investigation into the effect of an acutely-administered, intravesical TRPA1 agonist, CMA, on nociceptive responses to urinary bladder distension (UBD) in control and acutely bladder-inflamed rats; and 3. investigations similar to those of the second set of experiments but utilizing two doses of oral cinnamon in place of intravesical CMA.

### 2.2. Experiment 1: Effect of Bladder Inflammation on TRPA1 Content in Bladder

*Induction of Acute Cystitis.* Using a published protocol for inducing acute cystitis [25], adult (12–15 weeks of age) female rats were anesthetized with 2–5% isoflurane in oxygen, injected with ampicillin (50–100 mg/kg IP), their urethral orifice swabbed with an iodine-povidone solution, and a 22 gauge angiocatheter passed transurethrally into their bladder. A solution of Zymosan A (1% in normal saline; 0.5 mL; Sigma Aldrich, St. Louis, MO, USA) was injected into the bladder and allowed to dwell for 30 min, followed by passive draining. Rats were kept warm on a heating blanket, allowed to recover, and returned to their home cages. Control treatments consisted of a similar anesthetic for 30 min, iodine-povidone swabbing, ampicillin treatments, and identical recovery protocols. Treatments occurred only once and were performed approximately 24 h prior to tissue harvest or reflex testing described in subsequent sections.

*Enzyme-Linked ImmunoSorbent Assay (ELISA).* Twelve rats (n = 6/group) underwent induction of acute cystitis or control pretreatment according to the protocols described above. After 24 h, they were deeply anesthetized with 5% isoflurane and then euthanized via decapitation. Bladders were removed and processed according to ELISA kit instructions. Protein concentrations were determined using the Pierce BCA Assay Reagent Kit (Thermo Fisher Scientific, Rockford, IL, USA). TRPA1 content was quantified using an LSBio kit (Lifespan Biosciences, Seattle, WA, USA). Samples and serial dilutions of stan-

dards were processed according to kit protocols. A one-way ANOVA was used to identify treatment effects.

### 2.3. Experiment 2: Effect of Intravesical TRPA1 Agonist on Reflexes with & without Cystitis

*Surgical Preparation For Visceromotor Reflex Measures.* All rats received either a control or acute cystitis pretreatment 24 h prior to additional testing in a fashion identical to those of Experiment 1. On the day of testing and under isoflurane anesthesia (4% in oxygen), the right carotid artery was cannulated for recording of arterial blood pressure (sparing the vagus), thereby, monitoring the physiological status of the preparation. The trachea was cannulated for artificial ventilation. A 22-gauge angiocatheter (Johnson and Johnson, Arlington, TX, USA) was inserted into the urinary bladder via the urethra and held in place by a tight suture placed around the distal urethral orifice. Silver wire electromyographic (EMG) electrodes were placed into the left external oblique musculature for differential amplification and recording of EMG activity. Rats were not restrained in any fashion and body temperature was maintained using a heating pad. Isoflurane anesthesia was then reduced to approximately 0.75% in oxygen until animals manifested a withdrawal reflex to toe pinch and MAP stabilized. This typically required approximately 15 min. In two separate subgroups, either the mineral oil vehicle solution (VEH) or cinnamaldehyde (CMA–10% volume:volume in mineral oil; Source: Sigma Aldrich, St. Louis, MO, USA) was administered in a volume of 0.05 mL intravesically and allowed to dwell in place for 15 min. EMGs were continuously measured before, during, and after the infusion of the intravesical agent and averaged in one minute epochs so that responses to the administered agents could be determined. After these procedures, rats' EMG activity were recorded during testing with urinary bladder distension (UBD). Three 60 mmHg UBDs (20 s duration) were administered to overcome an initial period of bladder sensitization and were followed by a sequence of 20 sec duration graded UBDs at pressures of 10, 20, 30, 40, 50, 60, 70, and 80 mmHg, respectively (2 min ITI; performed in ascending order). In ten additional rats with acute cystitis (n = 5 in two groups), undergoing the same procedures, the TRPA1 antagonist HC030031 (0.05 mL, 10 mg/mL; Source: Cat# 2896, TOCRIS BioScience, Minneapolis, MN, USA; 50% DMSO vehicle) was infused, intravesically, 5 min prior to the administration of intravesical CMA or with no subsequent treatment. This was done to assess whether the observed effects of CMA were due to TRPA1 activation or due to nonspecific actions of the CMA.

*Study Protocol with Cystometrographic (CMG) measures.* A separate group of rats (n = 6/group) were anesthetized with a combination of urethane (1.4 g/kg IP; Sigma Aldrich, St. Louis, MO, USA) and isoflurane (initially 2–5% in oxygen for surgical preparation, then reduced to approximately ≤0.25%). All rats had received either a control or acute cystitis pretreatment 24 h prior to additional testing in a fashion identical to those of Experiment 1. Rats underwent surgical preparation identical to that described above, with the exception that vascular cannulation was not performed. After stabilization of anesthesia and following a 120 s pre-infusion period, either VEH or CMA (0.5 mL; 10% solution volume:volume in mineral oil vehicle) was infused into the bladder followed by a small air flush and the EMG measures recorded and averaged in one minute epochs so that responses to the administered agents could be determined. After 15 min, the bladder was drained and a slow infusion of normal saline was started using a CMG device (rate of infusion 50 μL/min; maximal infusion volume 1.0 mL) while continuous monitoring of the pressure inside the bladder was measured using an in-line, low volume pressure transducer. The volume infused when there was a sudden increase in intravesical pressure (indicating the first bladder contraction in response to filling) was recorded as the micturition threshold (MT). If no contraction occurred, a default value of 1.0 mL was utilized for statistical analysis.

*Data Analysis and Statistics.* EMGs were recorded during all procedures and when evoked, converted to "change" values by subtracting out the values of appropriate baselines measured from the period immediately preceding a given test. The evoked visceromotor response was defined as the mean mV of rectified EMG activity during the 20 s of UBD

minus the mean mV of rectified EMG activity measures during the 10 s prior to UBD. Statistics are presented as the mean ± S.E.M unless otherwise stated. Two-way repeated measures ANOVAs were used as appropriate. Post hoc contrasts were performed using Tukey's HSD with a family-wise $\alpha$ set at 0.05.

### 2.4. Experiment 3: Effect of Oral Cinnamon Administration in Rats with or without Cystitis

A set of experiments was performed similar to those described in Experiment 2 but instead of administering VEH and CMA intravesically, the rats were anesthetized 3 h prior to the start of the final experiment and gavaged with either normal saline (2 mL) or finely ground cinnamon power (100 or 400 mg/kg suspended in 2 mL normal saline). Rats were allowed to recover from anesthesia and then reanesthetized for a terminal experiment using a methodology identical to that described for Experiment 2 with a limitation of the preparation to the EMG measures evoked by UBD intensities of 10–60 mm Hg, and no vascular catheters or CMG measures were performed. The ground cinnamon employed was commercially available and derived from the bark of *Cinnamomum zeylanicum* (Syn C. verum family: Lauraceae; true cinnamon); Source: Ceylon Cinnamon Powder, SKU#669, My Spice Sage, Yonkers, NY, USA).

### 3. Results

### 3.1. Experiment 1: ELISAs

Zymosan-induced inflammation of the bladder resulted in a statistically significant increase in TRPA1 content in bladder tissues. Uninflamed bladder homogenates had 35.3 ± 2.5 pg/mL concentrations of TRPA1 versus inflamed bladders homogenates, which had 112.2 ± 32.0 pg/mL concentrations as measured by ELISA (n = 6 rats per group; ANOVA/unpaired *t*-test comparison $p = 0.035$).

### 3.2. Experiment 2A: Reflex Responses Immediately following Intravesical Drug Infusion

Figure 1 presents mean changes in EMG measured in one minute epochs for the first three minutes following the infusion of either VEH or CMA relative to the mean baseline activity obtained for 60 s prior to the infusions. Data from both sets of experiments (those subsequently measuring VMRs and those measuring CMGs) were combined. Notably, increases in EMG activity were statistically significant in the CMA-treated groups, when combined, compared with the VEH-treated groups (ANOVA analysis: $F_{1,1,61} = 5.433$, $p = 0.023$ for CMA treatment effect), but there did not appear to be a difference related to whether bladders were inflamed or not (ANOVA analysis: $F_{1,1,61} = 0.001$, $p = 0.992$).

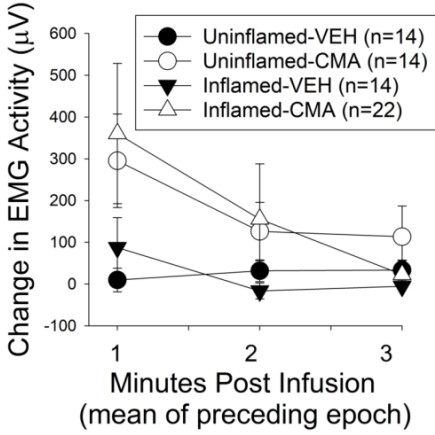

**Figure 1.** Response to intravesical infusion. Change in rectified EMG activity of superior oblique musculature when compared to baseline measures prior to drug infusion in rats with and without inflamed bladders. VEH indicates intravesical infusion of vehicle (mineral oil) whereas CMA indicates intravesical infusion of 10% cinnamaldehyde. There was a significant effect of CMA treatment but the presence/absence of zymosan-induced bladder inflammation did not have any statistically significant effect.

### 3.3. Experiment 2B: Responses to Graded Urinary Bladder Distension (UBD)

Figure 2A presents the EMG data obtained UBD testing following either VEH or CMA infusion. Differences did not appear until UBD pressures exceeded 20 mmHg. For graphical purposes, data for intensities $\geq$ 30 mm Hg are displayed but all data were utilized for statistical analysis. A surprising finding was that there was a statistically significant inhibitory effect of CMA vs. VEH treatment ($F_{1,1,37}$ = 4.374, $p$ = 0.043) and an interaction effect associated with the intensity of UBD and bladder inflammation status ($F_{7,7,259}$ = 2.168, $p$ = 0.037) as well as with CMA treatment ($F_{7,7,259}$ = 4.270, $p$ < 0.001). This was a finding directly opposite to our hypothesis. In Figure 2B, the intravesical administration of the TRPA1 antagonist HC030031 blocked the antinociceptive actions of subsequently administered CMA in rats with inflamed bladders (effect of HC treatment, repeated measures ANOVA $F_{1,22}$ = 9.360, $p$ = 0.006). By itself, intravesical HC030031 treatment also appeared to reduce nociceptive responses to UBD in rats with inflamed bladders subsequently treated with intravesical VEH, but not in a statistically significant fashion (ANOVA analysis: $F_{1,14}$ = 3.927, $p$ = 0.068).

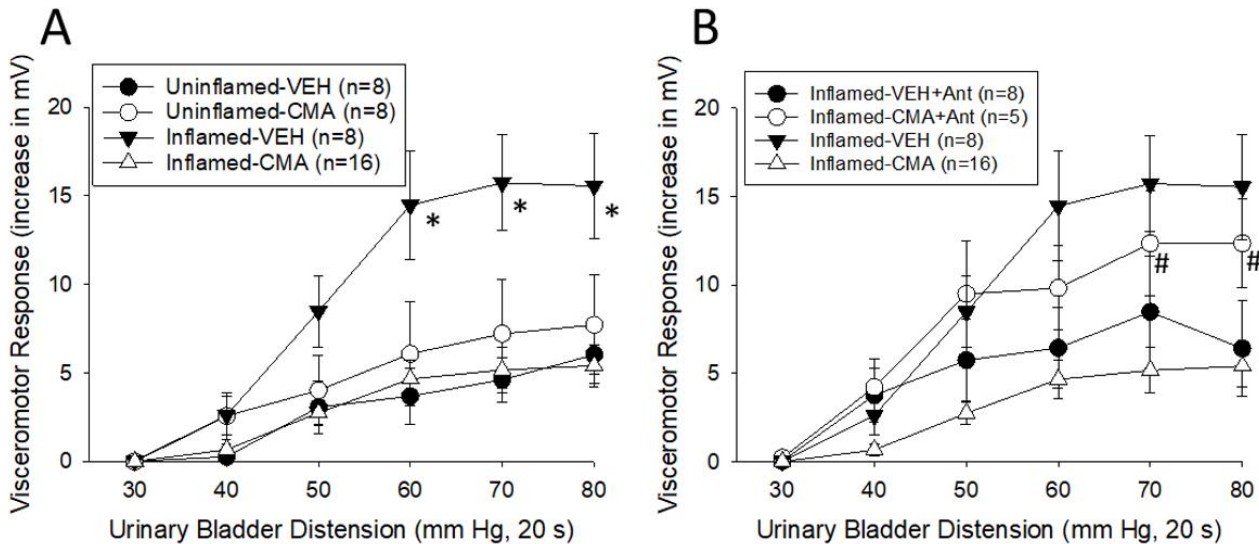

**Figure 2.** (**A**) Visceromotor responses to UBD in female rats with and without inflamed bladders after either intravesical vehicle (VEH) or cinnamaldehyde (CMA) infusion. * indicates a significant difference from all other groups ($p$ < 0.05). (**B**) Same data for rats with inflamed bladders as in (**A**) but, in addition, data from rats that also were pretreated with intravesical HC030031, a TRPA1 antagonist (indicated as +ANT). # indicates a significant effect of HC030031 pretreatment ($p$ < 0.05) at these intensities of UBD in CMA-treated groups.

### 3.4. Experiment 2C: Effect of Inflammation and Intravesical CMA on Cystometrograms

The amount of infusate needed to evoke an initial bladder contraction was quantified as the micturition threshold (MT) in microliters volume. A comparison of the different pretreatment groups demonstrated a statistically significant effect of CMA treatment in rats without inflamed bladders: the VEH group MT was 646 ± 66 µL whereas the CMA group MT was 278 ± 80 µL. Likewise, rats with inflamed bladders had statistically lower MTs in the VEH treatment group (MT was 205 ± 40 µL) when compared with the VEH treatment group of uninflamed rats, but there was no additional effect of CMA treatment in rats with inflamed bladders (MTs of 228 ± 21 µL). ANOVA analysis indicated the rats with uninflamed bladders treated with intravesical VEH were different from the other three groups (ANOVA analysis: $F_{3,20}$ = 9.273; $p$ < 0.001; pairwise post hoc comparisons using Tukey's HSD were all with $p$ < 0.01), but the three other groups did not differ from each other.

### 3.5. Experiment 3: Effect of Cinnamon on Visceromotor Responses to Graded Urinary Bladder Distension

A dose-dependent inhibition of the VMRs was noted due to oral cinnamon treatment in rats with inflamed bladders (Figure 3A). Consistent with the intravesical CMA data, a comparison of responses following oral saline with those following oral cinnamon at the 400 mg/kg dose in rats with inflamed bladders indicated a statistically significant *decrease* in activity due to the cinnamon treatment ($F_{1,9} = 8.256$, $p = 0.018$). The responses following 100 mg/kg of oral cinnamon were similar but did not reach statistical significance. In contrast to the intravesical CMA data, a dose-dependent *augmentation* of the VMRs was noted due to the oral cinnamon treatment in rats with uninflamed bladders (Figure 3B). A comparison of responses following oral saline with those following oral cinnamon at the 400 mg/kg dose in rats with uninflamed bladders indicated a statistically significant *increase* in activity due to the cinnamon treatment ($F_{1,8} = 6.641$, $p = 0.033$). The responses following 100 mg/kg of oral cinnamon did not differ from those in the oral saline-treated rats without bladder inflammation. This finding in rats without bladder inflammation was consistent with our hypothesis, but the clinical correlate to this group would be healthy individuals without the pain of cystitis.

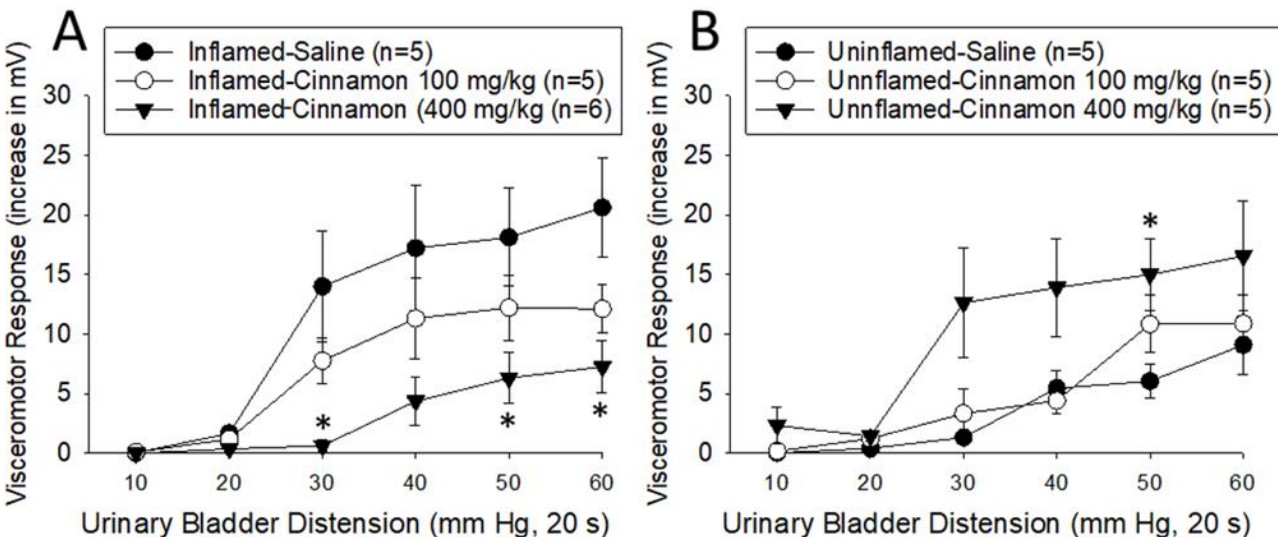

**Figure 3.** Dose-dependent effects of oral cinnamon on visceromotor responses to UBD in rats with inflamed bladders (**A**) and uninflamed bladders (**B**). An inhibitory effect was noted in rats with cystitis and an augmentation effect was noted in otherwise healthy rats. * indicates a significant difference from the saline-treated rats ($p < 0.05$).

### 4. Discussion

The most important finding of the present study was that TRPA1 agonists in the form of intravesical CMA or as oral cinnamon produced a reduction in nociceptive responses to distension of inflamed urinary bladders in female rats. In rats with uninflamed bladders, there was an insignificant effect of intravesical CMA on nociceptive responses but a statistically significant augmentation effect of cinnamon on VMRs at the highest dose. There was an excitatory effect of both bladder inflammation and CMA, together or independently, on bladder non-nociceptive reflexes such as those measured by CMG. Taken together, these data support that cinnamon ingestion could have an irritant effect in individuals without cystitis (which supports our hypothesis) but also produced a robust antinociceptive effect in the presence of acute cystitis suggesting a potential for use of cinnamon as a natural nutraceutical in the treatment of pain associated with cystitis. This second component of these studies was unexpected and does not support our original hypothesis.

Bladder inflammation has been demonstrated to induce increased TRPA1 agonist responsiveness in primary afferent neurons with endings in the bladder wall [16,26], an effect linked to growth factors such as artemin [15] and BDNF [27]. The present study also observed an increase in TRPA1 content within the bladder. This result, coupled with the present study's finding that administration of the TRPA1 antagonist HC030031 into the bladder prior to intravesical CMA administration blocked the inhibitory effect of the CMA in rats with inflamed bladders, further supports a TRPA1 mechanism that begins with the activation of urinary bladder primary afferent neurons. We have observed differences in subpopulations of bladder primary afferent neurons in relation to their TRPA1 agonist responsiveness [28]. There is also evidence from others for at least two different TRPA1 expressing neuronal groups associated with bladder sensation [29] including those that have different developmental triggers for expression of the receptor channel [30]. Altogether, these findings suggests that a particular neuronal subgroup with specific functions (such as activation of an antinociceptive system) may be selectively stimulated to generate increased expression of TRPA1 receptor channels. However, this does not rule out potential TRPA1-linked mechanisms acting through urothelial or other bladder substrates. These, in turn, may communicate with associated neural structures to produce reflex effects. Although the intravesical administration of a TRPA1 agonist selectively activates bladder primary afferent neurons, direct central effects, particularly in the case of oral cinnamon administration, are also possible. A case in point, Yamanaka et al. [31] observed that spinal inhibitory neurons had TRPA1 excitatory inputs. Similarly, oral drug administration also activates TRPA1 receptors in the mouth and gut, which could have secondary effects. Ingestion of food with the spice would likely affect that potential mechanism and so needs to be taken into account when considering clinical trials.

TRPA1 channels have been implicated in multiple urological functions (e.g., [32–34]) including responses to bladder infection [35,36]. Another TRP channel, TRPV1, already has an established role in a variety of bladder disorders. However, while vanilloid therapies have met with reports of success, its use has been generally discontinued due to either patient discomfort or failure to pass rigorous clinical trials [8]. This may be due, in part, to the clinical use of agonists to produce decoupling/desensitization of the TRPV1 channels rather than the more pharmacologically logical use of antagonists. In the case of TRPA1 ion channels, desensitization of that channel in cell cultures can occur by presentation of TRPA1 agonists, and this desensitization process requires multiple cellular pathways [1,2]. It is possible that the administration of TRPA1 agonists in the present study led to desensitization of primary afferents as opposed to activation of an inhibitory system.

All mechanistic issues aside, it is notable that these studies are potentially of high clinical impact and significance due to the ability to immediately translate components of these studies into clinical practice. The spice, *true cinnamon*, derived from cinnamon bark, is one of the oldest traditional medicines for inflammatory and pain-related disorders with an over 4000 year history of use [37,38]. Traditional medicine as well as internet-based experts (e.g., [39,40]) claim benefits from cinnamon as a treatment of arthritis and musculoskeletal disorders as well as a treatment for the symptoms of acute bladder infection. Patient reports include pain relief by some and pain exacerbation by others. These reports would appear consistent with the findings of the present study where opposite effects of cinnamon treatment were observed dependent on the presence or absence of active cystitis. Published preclinical (e.g., [41]), as well as clinical [42], studies give additional evidence for antinociceptive effects of cinnamon or cinnamon extracts. Pharmacokinetic studies have identified a significant amount of ingested cinnamon powder is excreted as TRPA1 agonists in the urine [43,44] with known effects on bladder function [44]. Due to its ready availability, it would seem that use of the spice would be the logical choice for initial clinical translation.

Given this ability of clinicians to immediately translate this information into practice, there is a compelling need for randomized controlled trials that will assess potential clinical efficacy of cinnamon ingestion on human bladder pain disorders such as acute bacterial

cystitis. If human studies are consistent with the present study, then, hopefully, clinicians will incorporate it into their practice as part of evidence-based medicine. With increasing problems associated with use and misuse of the current agents used for the treatment of pain, alternative therapies that have a favorable safety profile would be desirable.

## 5. Conclusions

We have demonstrated that TRPA1 agonist administration, including the use of oral cinnamon, has an inhibitory effect on bladder nociception in rats with inflamed bladders. Given the safety of these compounds, there is a compelling need for randomized controlled trials to test the efficacy of these agents on inflammatory bladder disorders.

**Author Contributions:** T.J.N. was involved in all aspects of this study from conceptualization to data collection, data analysis and writing; A.B. and M.E.N. were involved in data collection and writing; C.D. was involved in data collection, data analysis, data curation and writing. All authors have read and agreed to the published version of the manuscript.

**Funding:** These studies were supported by the United States National Institutes of Health grant R01DK51413.

**Institutional Review Board Statement:** These studies were approved by the University of Alabama at Birmingham Institutional Animal Care and Use Committee (IACUC-10250, app 10.30.2014; IACUC-22026, app. 4.7.2020).

**Data Availability Statement:** Raw data is available on request from the corresponding author.

**Acknowledgments:** Technical assistance was provided by Alan Randich.

**Conflicts of Interest:** None of the authors have any conflict of interest associated with the present manuscript.

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
