# Peer review of "TRPA1 Agonists and Bladder Nociception in Female Rats Suggest Potential for Nutraceutical Benefit from Cinnamon"

_nutraceuticals, doi:10.3390/nutraceuticals3010012_

Round 1

Reviewer 1 Report

This study shows that an achievement that analgesic potential of cinnamon for acute cystitis pain and will lead to practical applications. There were no problems in constructing the experimental strategy, and the results are clear. Although it is not enough to conduct additional experiments, the following points should be taken into consideration.

1.       Why did you use cinnamon powder for oral administration instead of cinnamaldehyde? In this study, the change in cinnamaldehyde concentration in the bladder is not clear for either experiment. It is also unknown how much orally administered cinnamaldehyde migrates into the bladder. It is also possible that it does not act directly on the bladder as the author suggests. To reduce the uncertainty factor, it might be necessary to use cinnamaldehyde for oral administration.

2.       The number of animals varies from 5 to 16. Is there a rationale for the necessary number?

3.       Oral administration in mice would be equivalent to ingestion as a capsule in humans.  If cinnamaldehyde is ingested as a food, its affection on TRPA1 in the oral cavity may also be affected and should be considered. Is the concentration of cinnamon powder in your experiment in the range used for food?

Author Response

1a.  We used cinnamon powder for oral administration in order to be most translational - this spice, available at most food markets, could be obtained easily by most subjects.  

1b.  We now indicate that a 10% cinnamaldehyde solution volume:volume in mineral oil vehicle was what was administered. 

1c.  We did not measure the actual urine concentrations of cinnamaldehyde in rats which received oral cinnamon, but others have described the pharmacokinetics of such compounds and demonstrated effects on bladder function of oral cinnamon powder consumptiion - we added references 43 and 44 which address these points.

2.  We have identified that n's of 5-8 are needed for robust pharmacological effects and so established those as the minimum number sampled.  The effect of the oral cinnamon was very robust.   The n=16 values were because data from both the VMR and CMG experiments both had the same measures performed during intravesical drug infusion so the n is a sum of both groups studied.

3.  The author has a good point that activation of TRPA1 receptors in the mouth (and we added gut) could contribute to effects -- we have now added a sentence to the manuscript with that idea. 

Reviewer 2 Report

The authors reported an interesting pharmacological study to investigate whether TRPA1 agonists produce pain or analgesic effect on naïve/acute zymosan-induced cystitis model. They found that CMA and cinnamon decreased VMRs to UDR compared with vehicle treatment in cystitis model rats. I have just a few comments listed below, which should help to improve the clarity of the contribution.

1. The resolution of each figure is not high enough in my reviewing PDF file. The labeled numbers of y-axis and labels of each group are difficult to read, especially in Figure.1.

2. Fig.2 Please explain the abbreviation of “Ant” in figure legend.

3. Fig.3 “Inflamed Cinnamon (400mg/kg (n=6)” should be “Inflamed-Cinnamon 400mg/kg (n=6)”.

4. The authors clearly showed the upregulation of TRPA1 channels in the bladder of model rats. However, the authors explained that the HC-030031 tended to reduce nociceptive responses to UBD in model rats. Does this mean that the TRPA1 channel itself does not contribute much to the generation of the nociceptive responses in this model?

Author Response

  1. We enlarged the inserted figures and will try to submit original TIFF files to give improved resolution
  2. We have added a definition of ANT to Figure 2
  3. We have added a hyphen to Figure 3
  4. We observed that HC-030031 attenuated the inflammation-related augmentation of visceromotor responses in vehicle-treated rats but not in a statistically significant fashion.  Given the reports by others (e.g., ref 16) we did not feel the need to further investigate this phenomenon and included so that it was apparent that the antagonist did not produce an augmenting effect by itself.  Instead, it suppressed the inhibitory effect of the CMA which is what we were trying to test.